# The Lack of Impact of Primary Care Units on Screening Services in Thailand and the Transition to Local Administrative Organization Policy

**DOI:** 10.3390/healthcare13151884

**Published:** 2025-08-01

**Authors:** Noppcha Singweratham, Jiruth Sriratanaban, Daoroong Komwong, Mano Maneechay, Pallop Siewchaisakul

**Affiliations:** 1Faculty of Public Health, Chiang Mai University, Chiang Mai 50200, Thailand; noppcha.s@cmu.ac.th; 2Department of Preventive and Social Medicine, Faculty of Medicine, Chulalongkorn University, Bangkok 10330, Thailand; sjiruth@gmail.com; 3Sirindhorn College of Public Health Phitsanulok, Faculty of Public Health and Allied Health Sciences, Praboromarajchanok Institute, Phitsanulok 65130, Thailand; daoroong.k@scphpl.ac.th; 4Banmi District Public Health Office, Lopburi 15000, Thailand; drmano2558@gmail.com

**Keywords:** primary care unit (PCU), local administrative organization (LAO), health service delivery, diabetes screening, hypertension screening, generalized estimating equations (GEE), health policy evaluation, Thailand health system

## Abstract

**Background/Objectives**: In Thailand, the transition of primary care units (PCUs) to Local Administrative Organizations (LAOs) has raised concerns regarding the potential impact on healthcare service delivery. This study aimed to compare health services between PCUs that have been transferred to LAOs and those that have not. **Methods**: A total of 15 transferred PCUs (T-PCUs) and 45 non-transferred PCUs (NT-PCUs), matched by population within the same provinces, were purposively sampled. The study population consisted of the cumulative number of diabetes (DM) and hypertension (HTN) screenings retrieved from the National Health Security Office (NHSO) database from 2017 to 2023. The impact of the LAO transfer policy on health service delivery was assessed using generalized estimating equation (GEE) models. All analyses were performed using Stata version 15. **Results**: The result showed no significant difference in the population and size of PCUs. DM screening was non-significantly lower by 18.9% (AdjRR: 0.811), and HTN screening was lower by 18.6% (AdjRR: 0.814), when comparing T-PCU with NT-PCU. Similarly, the DM and HTN screening in T-PCU was non-significantly lower than NT-PCU when interacting with time. Both T-PCU and NT-PCU show decreases over time; however, the decrease was not statistically significant. **Conclusions**: Our results show a non-significant difference in DM and HTN screening between T-PCU and NT-PCU. Therefore, decentralization did not clearly demonstrate a negative impact on the delivery of these health services. Further research is needed to consider other confounding and covariate factors for DM and HTN screening.

## 1. Introduction

Improving health through clinical and public health services is one of the primary objectives of any healthcare system in the world [1]. The world health organization has developed a framework of “six building blocks”, which are key components needed to strengthen health systems, including service delivery, health workforce, health information systems, access to medicines, vaccines and technologies and health financing and leadership and governance [2]. A primary care unit (PCU) plays a vital role across all six building blocks of a health system, as it is often the first point of contact for individuals in the health system. Thus, PCUs are the core of service delivery.

PCU is one of the mechanisms of healthcare systems which focuses on health at all ages. It involves prevention, health promotion, treatment, rehabilitation, and palliative care. This approach empowers people and communities to choose healthier lifestyles, prevent diseases, and access early detection, treatment, and recovery services. It guarantees that individuals will receive assistance in more accessible settings and that they will be referred to secondary or tertiary care establishments, such as hospitals, if needed [3]. This is similar to the system used in Thailand, where primary care serves as the first point of access to healthcare [4].

In Thailand, primary healthcare forms the foundation of the healthcare system, with a strong emphasis on universal access, prevention, health promotion, and early detection of diseases. Its services are primarily delivered through Sub-district Health-Promoting Hospitals (SHPHs), which are equivalent to PCUs. These were previously referred to as Tambon Health Promotion Centers or Tambon Primary Care Units and serve as the first point of contact for individuals and communities, particularly in rural areas. These facilities provide essential health services, including maternal and child healthcare, immunizations, management of common illnesses, health education, and screening services for non-communicable diseases (NCDs) such as hypertension, diabetes, and cervical cancer [5,6]. Screening services aim to promote early detection and timely management, which aligns with Thailand’s national health policies focused on strengthening community-based care and reducing health disparities [5]. Understanding this PCU structure is essential for contextualizing the delivery of health services and assessing the performance of PCUs.

Non-communicable diseases (NCDs) account for more than two-thirds of global deaths, with a rising burden of disease [7]. Diabetes (DM) and hypertension (HTNN) are the two main drivers of NCD prevalence globally, including in Thailand [8,9,10,11]. In countries with limited resources, adapting disease guidelines have meant that non-physician clinicians are required to deliver care and to ensure effective implementation of standardized protocols for diagnosis, treatment, and monitoring [12]. Previously, patients with DM/HTNN relied on hospitals for treatment, which was burdensome for the hospitals’ service capacity. To mitigate this burden and to increase access to care, in 2002, universal health coverage (UHC) was launched, and has since made good progress [4]. UHC is not only beneficial to the Thai population in terms of healthcare coverage but also in sustaining engagement with primary care services. Under their benefits package, Thai citizens receive personal health-promoting and disease prevention services free of charge, especially for non-communicable diseases (NCDs). One of the most important roles of primary care units is to screen the DM/HTN population in their jurisdiction every year, as required by the benefit package of the National Health Security Office (NSHO). However, the challenges of chronic care include a health system that incorporates adequate service components to provide quality of care [13].

Despite the presence of UHC in Thailand over the past two decades, provision of chronic care for patients with HTN and/or DM in primary care settings has faced challenges related to a shortage of healthcare workers, particularly nurses [14]. To improve efficiency and quality and ensure that the healthcare service would remain both equitable and effective [15], decentralization was introduced as a reform measure. In 2021, one of the major policy agendas of the health system in Thailand was the transfer of PCUs from the Ministry of Public Health to the governance of the PAO. These PCUs are referred to as transferred PCUs (T-PCUs). As of July 2022, 4275 PCUs (43.5%) had been transferred [16]. Therefore, in Thailand, there are two classifications of PCUs based on administrative governance. T-PCUs refer to PCUs that have been administratively transferred to operate under Provincial Administrative Organizations (PAOs) as part of the decentralization policy. In contrast, non-transferred primary care units (NT-PCUs) remain under the direct management of the Ministry of Public Health (MOPH).

Studies have shown that decentralization increases efficiency by empowering local providers and communities. It can lead to more effective resource allocation and improved service delivery [17]. In contrast, a review study revealed that decentralization has diverse effects on the core fundamental components of the health system—both positive and negative—but overall, the effects tend to be more negative. Another study also suggests that research should focus on specific aspects of decentralization and health system concepts within particular setting [18].

Although the transfer is still in progress, there are several challenges that have arisen due to the fact that they might not be entirely consistent with the objectives or requirements of each party, both conceptually and practically. A recent report on the results of a first-phase assessment of population health impacts following the transfer of PCUs to PAOs showed changes in the provision of services at PCU, especially in resource management and in the relationships between agencies and personnel teams in areas where PCUs have become PAOs [19]. Furthermore, previous research has investigated the impact on governance, management, and financial aspects [20], but few have investigated the impact on health service outcome [21]. As a result, these changes might affect DM and HTN screening services under PCUs in Thailand.

Fifty-eight HTN primary care units were turned into LAOs between 2007 and 2022. This enables us to make a comparison between transferred PCUs and non-transferred PCUs that are in a stable stage, having adapted to the policy and overcome the transition state. Furthermore, as screening has been implemented annually, it is also beneficial to observe the changes in DM and HTN screening services over time. Therefore, this study hypothesized that there would be no significant difference between the NT-PUC and TPUC in DM and HTNN screening service.

Based on the above statements, this study aims to investigate the impact of the policy transferring PCUs to the PAO policy on health, with a focus on DM and HTNN screening services in Thailand.

## 2. Materials and Methods

### 2.1. Study Design

This study employed a retrospective matched-pair cohort design to compare outcomes between transferred and non-transferred primary care units (PCUs) in Thailand. Each transferred PCU (T-PCU) was matched with three non-transferred PCUs (NT-PCUs) based on population size and location within the same province to ensure comparability. This matching strategy was used to minimize confounding and enhance the validity of comparisons of the service performance between the two groups.

### 2.2. Population and Sampling

The population consisted of NT-PCUs and T-PCUs in Thailand, with a total of 9836 PCUs. However, in this study, T-PCUs are the PCUs that became LAOs between 2007 and 2022, totaling 58 T-PCUs. In this study, the sample was selected based on the following inclusion criteria:Transferred between 2007 to 2012.Complete screening data.These criteria were used to ensure that the selected facilities had already undergone the transition period and made organizational adjustments following the transfer policy. Therefore, a total of 15 T-PCUs were selected.

### 2.3. Data Source

To increase statistical power in this study, 15 T-HPHs were purposively recruited along with 45 NT-HPHs matched by population within the same provinces. Hypertension and diabetes were defined according to the ICD-10 codes E10 to E14 and I10 to I15, respectively. The hospital ID and number of hypertension and diabetes screenings were retrieved from the National Health Security Office database for the period between 2017 to 2023.

The outcome of this study was screening, defined as the number of people who were screened at their registered PCU.

### 2.4. Statistical Analysis

Categorical data were reported as numbers and percentages, and continuous data were summarized in the form of means with standard deviations or medians with ranges (minimum–maximum). The proportions different according to population and size of T-PCU and NT-PCU were tested using Chi-square test and Fisher’s exact test.

Since the dependent variable in this study is the DM and HTN screening number measured across consecutive years, and the data are repeated measures within the same PCU, the generalized estimating equation model (GEE) was used to account for within-unit correlation over time. Both Poisson and negative binomial distributions were tested to assess overdispersion, and model selection was based on the Quasi-likelihood under the Independence model Criterion (QIC). The final GEE model, using a negative binomial distribution and an exchangeable correlation structure, was employed to estimate the association between PCU-related factors and screening numbers. The magnitude of the effects of PCU was reported as the adjusted Rate Ratio (adjRR) and their 95% CI. Statistical significance was defined as *p* < 0.05. All analyses were conducted using Stata version 15.

### 2.5. Ethical Approval

The study was reviewed and approved by the Institute for the Development of Human Research Protections (IHRP No. 111-2565).

## 3. Results

Table 1 presents the characteristics of transferred (T-PCUs) and non-transferred primary care units (NT-PCUs). The study included 15 T-PCUs and 45 NT-PCUs. The median population size served by T-PCUs was 4521 (IQR: 4549), compared to 4536 (IQR: 3679) for NT-PCUs, with no statistically significant difference (*p* = 0.79). Regarding facility size, 3 of the T-PCUs were classified as small, 10 as medium, and 2 as large, while among the NT-PCUs, 12 were small, 27 medium, and 6 large. The distribution of facility size did not differ significantly between the two groups (*p* = 0.91).

Table 2 and Figure 1 display the mean number of individuals screened for diabetes and hypertension between 2017 and 2023.

In total, 653,952 and 646,180 individuals were screened for DM and HTN, respectively, under T-PCUs. In comparison, 162,996 and 161,316 individuals were screened for DM and HTN under NT-PCUs. It is important to note that NT-PCUs were selected at a 3:1 ratio to each T-PCU in this study.

Over the entire observation period, NT-PCUs consistently reported higher average screening volumes than T-PCUs. The highest average screening for both conditions occurred in 2019, while the lowest was observed in 2022. In 2023, the largest gap between NT-PCUs and T-PCUs was recorded. The smallest difference in screening numbers occurred in 2020.

The effect of T-PCU on screening for diabetes showed that T-PCUs had a non-significantly lower screening rate of 0.811 (95% CI: 0.468–1.403) compared with NT-PCUs. An increase in years showed a non-significant decrease in screening by 2.2% (adjRR = 0.978; 95% CI: 0.953–1.01). T-PCUs, with increasing years, had a non-significantly (2.8%) lower screening rate (adjRR = 0.972; 95% CI: 0.93 to 1.024) compared with NT-PCUs (Table 3).

A similar phenomenon was also observed in the screening of hypertension, the effect of T-PCU on hypertension screening showed that T-PCUs had a non-significantly lower screening rate (0.814 lower (95% CI: 0.47–1.41)) compared with NT-PCUs. The increase in years showed a non-significant (2.3%) decrease in screening (adjRR = 0.977; 95% CI: 0.951 to 1.01). T-PCUs, with increasing years, had a non-significant (2.9%) reduction in screening (adjRR = 0.971; 95% CI: 0.922 to 1.023) compared with NT-PCUs (Table 3).

Table 4 shows the impact of time on diabetes and hypertension screening. We found that the number of screenings conducted in both PCUs decreased over time; however, the decrease was not statistically significant.

## 4. Discussion

A tremendous change in the public health system in Thailand has occurred in recent years: PCUs have been transferred to PAOs. Most related studies have focused on budget allocation; however, a few have investigated the impact on health services. Thus, this study investigated the number of screenings for HTN and DM conducted in T-PCUs and NT-PCUs. The DM and HTN screening rates in T-PCUs were lower than those in NT-PCUs, though the differences were not statistically significant. Screening rates in both groups declined over time; however, these trends did not reach statistical significance, including the interaction with time.

The previous evidence revealed various problems following the transfer of PCUs to LAOs, including healthcare service provision, disease prevention, and healthcare information management [22]. Based on the study’s findings, from 2017 to 2023, DM and HTN screening in T-PCU was non-significantly lower than NT-PCU alone, including when interacting with time. This aligns with evaluations of the transfer of primary healthcare services, which showed a similar trend to other health services with regard to screening for diabetes, hypertension, and cervical cancer [23]. A previous study conducted in Thailand, which focused on dental services, found that T-PCUs generally provided fewer dental services than NT-PCUs. In the case of dental care, the primary reason was that the transfer of responsibilities created uncertainty regarding job roles and caused apprehension among some dental public health officers. Furthermore, there was a lack of systematic support from parent hospitals in terms of dentists, equipment, instruments, medications, and non-medical supplies [21]. A study by Sriratanaban et al. also reported that LAOs placed less emphasis on health because this responsibility had only recently been shifted to them [19]. This resulted in disparities between transferred and non-transferred SHPHs [24]. However, DM and HTN screenings do not require specialized equipment or advanced healthcare staff expertise. Therefore, the difference in screening numbers was not statistically significant in our study.

Another possibility is that the village health volunteer (VHV) is considered to be an intermediate communicator between healthcare professionals and people in the community [25] and plays a crucial role in monitoring these NCDs, which are still under the MPH. Therefore, communication in terms of promotion and prevention (PP) programs might be interrupted by this working mechanism. However, a study by Sriratanaban et al. shows a good relationship between VHV and PCU, as they have been working together and known each other well over a long time. Thus, it would be easy to ask for cooperation and implementation of a PP program. It is very interesting to consider whether, in the long run, when healthcare professionals, VHVs, and other staff are replaced by new personnel, their relationships and cooperation will remain strong [19]. Hence, future research needs to collect empirical data which can be used to investigate the role of VHV as a mediating factor.

Notably, regardless of the transfer or non-transfer of the service area PCUs, we found that screenings for both conditions decreased over time. Recently, policy-makers have focused on treatment rather than screening, because reimbursement for screening is not the main payment method for promoting health promotion and prevention at the community level. NHSO-allocated CUP does not distribute the budget for screening because the hospital already provides the screening tools, i.e., for DM they provide needles so they are not required to give the money to the PCU. Therefore, this may explain the decrease in screening services in both transferred and non-transferred PCU [19]. Furthermore, the growth and expansion of cities into urban areas play pivotal roles in shaping public health, particularly in relation to transportation [26]. This may lead to an increase in the number of people who prefer accessing district hospitals rather than primary care units (PCUs), where medical doctors are often unavailable.

Decentralized health system governance can also have adverse effects. For example, less well-resourced local governments may struggle to generate sufficient revenue, leading to inadequate funding allocations. In the absence of centrally funded vertical health programs, this may result in insufficient support for preventive services [27]. A further concern pertains to the fact that managing a decentralized system requires effective coordination mechanisms between different levels of government and healthcare providers to ensure that services are integrated and efficient. As noted in a study from Pakistan, despite increased provincial health allocations, a lack of coordination between provincial and central levels impeded effective implementation [28]. Ongoing leadership turnover at the provincial level may have further disrupted coordination efforts [29]. Therefore, in the future round of PCU transfers, it would be challenging for PCUs to sustain coordination and funding resources to continue providing its service to their population.

It is inevitable that this study has some limitations. This study investigated only the selected T-PCU transferred between 2007 and 2012, with only 15 T-PCU. Therefore, the generalizability of our results is limited. There are many covariates and possible confounding factors that we did not consider, as our data were aggregated and only a few variables were available. There was also a limitation in the selection of the health index, which included only DM and HTN in primary care units. Furthermore, this study used claim data, which may not represent the real number of screening services.

Regardless of the limitations, this is the first study to investigate the impact of the transferred policy on screening services, based on the first round of PCU transfers using NHSO data. Though there may be concerns regarding the use of reimbursement data, the NHSO reimburses screening services based on a fee schedule. The reimbursement codes are the same; the NHSO does not distinguish between T-PCUs and NT-PCUs. This reimbursement method may encourage PCUs to provide services more intensively, thereby reducing the likelihood of underestimating the actual number of screenings.

## 5. Conclusions

Our results show a non-significant difference in DM and HTN screening between T-PCU and NT-PCU. This implies that decentralization did not clearly demonstrate a negative impact on the delivery of these health services. In future rounds of transfers, PCUs should continue to provide similar screening and other health services. Further research is needed to consider other confounding and covariate factors influencing DM and HTN screening. Furthermore, qualitative studies are needed to explore the details of the health service system during the transfer policy in Thailand.

## Figures and Tables

**Figure 1 healthcare-13-01884-f001:**
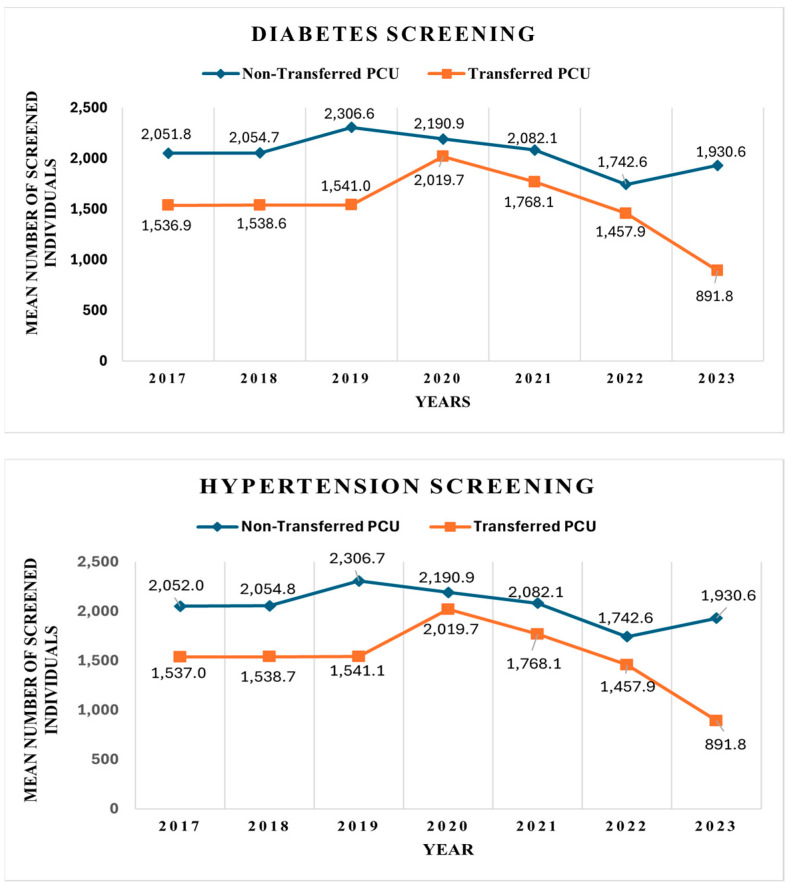
Trends in the mean number of individuals screened for diabetes and hypertension, 2017–2023.

**Table 1 healthcare-13-01884-t001:** Transferred and non-transferred PCU characteristics.

	T-PCU	NT-PCU	*p*-Value
Population, Median (IQR)	4521 (4549)	4536 (3679)	0.79 *
Size of PCU			
Small	3	12	0.91 **
Medium	10	27	
Large *	2	6	

* Chi-square test; ** Fisher’s exact test; T-PCU: transferred primary care unit; NT-PCU: non-transferred primary care unit.

**Table 2 healthcare-13-01884-t002:** Mean number of individuals screened from 2017–2023.

**Diabetes**				**Hypertension**			
**Year**	**NT-PCU**	**T-PCU**	**NT-PCU vs. T-PCU**	**Year**	**NT-PCU**	**T-PCU**	**NT-PCU vs. T-PCU**
2017	2051.82	1536.9	514.92	2017	2052.0	1537.0	515.0
2018	2054.68	1538.63	516.05	2018	2054.8	1538.7	516.1
2019	2306.55	1540.97	765.58	2019	2306.7	1541.1	765.6
2020	2190.87	2019.73	171.14	2020	2190.9	2019.7	171.1
2021	2082.09	1768.13	313.96	2021	2082.1	1768.1	314.0
2022	1742.56	1457.93	284.63	2022	1742.6	1457.9	284.6
2023	1930.58	891.8	1038.78	2023	1930.6	891.8	1038.8

T-PCU: transferred primary care unit; NT-PCU: non-transferred primary care unit.

**Table 3 healthcare-13-01884-t003:** Impact of transferring PCU on screening for diabetes and hypertension.

Variables	Diabetes	*p*-Value	Hypertension	*p*-Value
	AdjRR (95% CI)		AdjRR (95% CI)	
T-PCU (vs NT-PCU)	0.811 (0.468–1.403)	0.453	0.814 (0.47–1.41)	0.462
Years	0.978 (0.953–1.0)	0.096	0.977 (0.951–1.01)	0.100
T-PCU ## Years	0.972 (0.93–1.024)	0.290	0.971 (0.922–1.023)	0.278

AdjRR: adjusted rate ratio; T-PCU: transferred primary care unit; NT-PCU: non-transferred primary care unit.

**Table 4 healthcare-13-01884-t004:** Impact of time on diabetes and hypertension screening performed by the transferred primary care unit.

Variables	Screening	
	AdjRR (95% CI)	*p*-Value
**Diabetes**		
T-PCU Years	0.951 (0.900 to 1.005)	0.076
NT-PCU Years	0.978 (0.955 to 1.002)	0.074
**Hypertension**		
T-PCU Years	0.950 (0.891 to 1.004)	0.073
NT-PCU Years	0.977 (0.953 to 1.005)	0.078

AdjRR: adjusted rate ratio; T-PCU: transferred primary care unit; NT-PCU: non-transferred primary care unit.

## Data Availability

The data that support the findings of this study are available from the corresponding author upon reasonable request.

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
