# Peer review of "The Lack of Impact of Primary Care Units on Screening Services in Thailand and the Transition to Local Administrative Organization Policy"

_healthcare, 2025, doi:10.3390/healthcare13151884_

Round 1
Reviewer 1 Report
Comments and Suggestions for Authors
- Inclusion of only the 15 T-PCUs transferred in 2012, with no justification given for the exclusion of the remaining PCUs.
-
In the discussion section, the authors do not provide any qualitative evidence, such as interviews or relevant data, suggesting that additional qualitative evidence explains the mechanism.
-
It is recommended to supplement the review and explanation of primary health care and screening services to clarify the conceptual context.
-
The conclusion of the article is unclear and does not summarize the policy impact.
-
Are the quantitative methods used for data analysis overly complex? It is recommended to add a theoretical framework to make the analysis and conclusions clearer.
-
The NHSO reimbursement database was used to calculate the number of people screened, but the following issues were not discussed:
Are screening services covered by reimbursement? If they are free services, the data may underestimate the actual number of screenings.
Are there differences in reimbursement codes between T-PCU and NT-PCU?
Author Response
Response to reviewer
Thank you very much for your fruitful comments. We revised as much as possible. We hope that it will satisfy the reviewer. However, please let us know if any need to be edited.
Reviewer 1
- Inclusion of only the 15 T-PCUs transferred in 2012, with no justification given for the exclusion of the remaining PCUs.
Response: Thank you very much for your comments. We have extended our justification in the 2.2 Population and sampling and 2.3 Data source section as follows
“2.2. Population and sampling
The population were NT-PCU and T-PCU in of Thailand, total was 9836 PCU. However, in this study, the T-PCU was the PCUs that have been moved to LAO between 2007 to 2022, in which there were total of 58 T-PCU. In this study, the sample were included as follows the inclusion criteria of
1 transferred between 2007 to 2012
2 complete screening data
These criteria were used to ensure that these facilities had already undergone the transition period and organizational adjustments following the transfer policy. Therefore, the total of 15 T-PCU were selected.
2.3. Data source
To increase statistical power in this study, this study therefore purposively recruits the 15 T-HPH sample of 45 NT-HPH matched to their population within the same prov-inces. Hypertension and diabetes were defined according to ICD-10 code of E10 to E14 and I10 to I15. The hospital id, number of screenings of hypertension, and diabetes re-trieved from the National Health Security Office data-based between 2017 to 2023.
The outcome of this study is screening. Screening number was determined as number of people who have been screened in their registered PCU.”
(Line 120-138)
- In the discussion section, the authors do not provide any qualitative evidence, such as interviews or relevant data, suggesting that additional qualitative evidence explains the mechanism.
Response: Agreed. Thank you very much for the suggestions. This study unfortunately did not have the qualitative study to fulfil the in-depth information for discussion. However, we have put this in the last part of the conclusion, recommend future study needs to implement qualitative study for an in-dept understanding. (Line 274-280)
- It is recommended to supplement the review and explanation of primary health care and screening services to clarify the conceptual context.
Response: Thank you very much for this point. We have revised the introduction and tailored the rationale, which begins with the 6-building box, primary health care is part of the block which major core in health service, links to primary care unit and its responsibility. We also point out there were many services, but the most common one is screening NCD (DM and HT). (Line 38-64)
- The conclusion of the article is unclear and does not summarize the policy impact.
Response: As we revised our analysis based on other reviewers' suggestions, the results show no impact (i.e., no statistically significant difference or decrease). Therefore, we have updated the conclusion accordingly and emphasized that the T-PCU policy has no impact on these services. The revised conclusion was as follows
“Our results show a non-significant difference in DM and HT screening between T-PCU and NT-PCU. The T-PCUs to LAOs did not appear to impact the delivery of these health ser-vices. Therefore, the previous transfer PCU to LAOs show non-negative in health service system. Future research is needed to explore long-term effects with larger sample size and other service dimensions beyond DM and HT screening. Furthermore, the qualitative study is needed to explore in detail of health service system during the transfer policy in Thailand.” (Line 274-280)
- Are the quantitative methods used for data analysis overly complex? It is recommended to add a theoretical framework to make the analysis and conclusions clearer.
Response: Thank you very much for the recommendation. We add the extent of the reason/ theoretical of using the GEE in the data analysis part. GEE was selected for it could deal with repeated outcome which the outcome in this study is screening number with repeated year. (Line 144-153)
As another reviewer suggests bringing up the service delivery therefore, our analysis just focus on the screening service for DM and HT which is part of the 6 building blocks. (Line 38-45)
- The NHSO reimbursement database was used to calculate the number of people screened, but the following issues were not discussed:
Are screening services covered by reimbursement? If they are free services, the data may underestimate the actual number of screenings.
Response: Thank you very much for your suggestions. We have added in the discussion part. The NHSO reimburses screening services based on a fee schedule. This reimbursement method may encourage PCCs to provide services more intensively, thereby reducing the likelihood of underestimating the actual number of screenings.
(Line 246-250)
Are there differences in reimbursement codes between T-PCU and NT-PCU?
Response: Thank you very much for your suggestions. We have added in the discussion part.The reimbursement codes are the same; the NHSO does not distinguish between T-PCUs and NT-PCUs. (Line 246-250)
Please note that based on our new results. Therefore, we change the title to "No Impact of Primary Care Unit on Screening Services in Thailand: Transferred to the Local Administrative Organization Policy"
Reviewer 2 Report
Comments and Suggestions for Authors
The manuscript examines the impact of Thailand’s decentralization policy specifically, the transfer of Primary Care Units (PCUs) to Local Administrative Organizations (LAOs) on diabetes and hypertension screening rates. This topic is timely and relevant to both Thai health policy and broader global health system reform discussions. However, while the manuscript makes an initial contribution, several scientific, methodological, and editorial shortcomings significantly undermine its clarity, validity, and generalizability.
First,
-
The manuscript does not articulate its novelty clearly. Decentralization in health care has been extensively studied globally and to some extent within Thailand. But all this is missing in this manuscipt. The researchers should revisit the introduction part as well as the discussion part and include the relevant articles, and reframe the writeup.
-
The policy relevance is implied but not systematically framed. The authors should connect their findings more directly to current debates in Thai healthcare governance.
-
The study would benefit from discussing how this evidence can inform future rounds of PCU transfer or performance monitoring mechanisms.
Major Issues:
-
The matching methodology is insufficiently described. The authors mention “purposive sampling” but also describe matching by population size. Were other covariates considered (e.g., rural/urban, staffing, geographic disparities)? This is crucial given the small sample size (15 T-PCUs).
-
The sample size (15 T-PCUs and 45 matched NT-PCUs) is small and likely underpowered. While limitations are briefly acknowledged, the authors should explicitly address statistical power and potential selection bias.
-
Key confounders are unmeasured. For example, were there differences in staffing levels, funding allocation, community health volunteer (VHV) engagement, or physician availability between T- and NT-PCUs?
-
The use of GEE with Poisson distribution is appropriate in theory, but model diagnostics, assumptions (e.g., overdispersion), and robustness checks are missing.
-
The variable “time” is treated linearly across 7 years; however, no information is given on how time trends or external health system changes (e.g., COVID-19) may have affected screening behavior.
-
The interpretation of coefficients (e.g., -0.225) lacks practical context. How many fewer screenings does this represent in real terms?
-
Claims such as “twice as low” are vague and imprecise. Use percentages or rate ratios.
-
The presentation would benefit from graphical trends over time for T-PCU and NT-PCU, which would visually convey the temporal pattern more effectively.
The discussion is speculative in parts and lacks depth in comparing findings with international literature. For example:
-
-
How do the findings compare with decentralization experiences in other LMICs?
-
Could the reduced screening be partially due to non-policy externalities, such as urban migration or shifting patient preferences?
-
-
The role of VHV is mentioned but not empirically assessed. If VHV engagement is a hypothesized mediating factor, that should be explicitly modeled or at least flagged for future research.
The manuscript contains numerous grammatical, syntactic, and idiomatic errors. Some examples:
-
“...T-PCU has twice as low screening...” → unclear and ungrammatical.
-
“...was concluded, for they are in different sectors.” → confusing phrasing.
-
“The screening was decreasing twice...” → needs rewording.
-
The abstract is repetitive and grammatically weak. Avoid redundancy and clarify the methodology/results with concise, polished language.
-
The introduction includes policy background but lacks theoretical framing. Include references to relevant health policy or systems frameworks (e.g., WHO's health system building blocks).
While the topic is important and the data potentially valuable, substantial work is needed to bring this manuscript up to the publication standards of a peer-reviewed international journal. I look forward to reviewing a revised version that addresses the above concerns in detail.
Author Response
Response to reviewer
Thank you very much for your fruitful comments. We revised as much as possible. We hope that it will satisfy the reviewer. However, please let us know if any need to be edited.
Reviewer 2
The manuscript examines the impact of Thailand’s decentralization policy specifically, the transfer of Primary Care Units (PCUs) to Local Administrative Organizations (LAOs) on diabetes and hypertension screening rates. This topic is timely and relevant to both Thai health policy and broader global health system reform discussions. However, while the manuscript makes an initial contribution, several scientific, methodological, and editorial shortcomings significantly undermine its clarity, validity, and generalizability.
First,
1 The manuscript does not articulate its novelty clearly. Decentralization in health care has been extensively studied globally and to some extent within Thailand. But all this is missing in this manuscipt. The researchers should revisit the introduction part as well as the discussion part and include the relevant articles, and reframe the writeup.
Response: Thank you very much. We agree and have revised the introduction (Line 90-106) and discussion integrated with international study.
- The policy relevance is implied but not systematically framed. The authors should connect their findings more directly to current debates in Thai healthcare governance.
Response: We have reframed the introduction and added the Thai healthcare study in the introduction (whole introduction).
- The study would benefit from discussing how this evidence can inform future rounds of PCU transfer or performance monitoring mechanisms.
Response: As our results show no impact of PCU on screening service, which implies that there was no different the screening service between NT-PUC and T-PCU. Therefore, based on the current findings, the future round of PCU transfer should provide similar screening and other health services.
Major Issues:
- The matching methodology is insufficiently described. The authors mention “purposive sampling” but also describe matching by population size. Were other covariates considered (e.g., rural/urban, staffing, geographic disparities)? This is crucial given the small sample size (15 T-PCUs).
Response: Thank you very much for your comments. We selected just these 15-PCU with the purpose of reducing the confounding of the instant transition period and also the available of the data. We did revise and emphasize it as follows
“ The population were NT-PCU and T-PCU in of Thailand, total was 9836 PCU. However, in this study, the T-PCU was the PCUs that have been moved to LAO between 2007 to 2022, in which there were total of 58 T-PCU. In this study, the sample were in-cluded as follows the inclusion criteria of
1 transferred between 2007 to 2012
2 complete screening data
These criteria were used to ensure that these facilities had already undergone the transition period and organizational adjustments following the transfer policy. Therefore, the total of 15 T-PCU were selected.” (Line 120-138)
Furthermore, we totally agreed with reviewer concern of non-adjusting with rural/urban, staffing, geographic disparities factors and so on. As for the limitation of data, we have only aggregate data and few variables are available, so we could not take any covariates into consideration. However, we did put this issue in limitations. (Line 251-259)
we did also suggest for future research at the conclusion section (Line 274-279)
- The sample size (15 T-PCUs and 45 matched NT-PCUs) is small and likely underpowered. While limitations are briefly acknowledged, the authors should explicitly address statistical power and potential selection bias.
Response: Thank you very much for your fruitful comments. As of the purpose T-PCU that has just only 15-PCU available data therefore we match with 1:3 ratio to improve the power. Regarding the power of test diagnostic, we spent time finding the way to calculate it, but we sincerely state that we can’t find the way to perform the power of GEE when applying with count data. However, we could see the 95%CI that demonstrate not too wide CI which indicates more precise estimates, often due to larger sample size or less variability.
6 Key confounders are unmeasured. For example, were there differences in staffing levels, funding allocation, community health volunteer (VHV) engagement, or physician availability between T- and NT-PCUs?
Response: Thank you very much for your concern. We agree with these important unmeasured confounding factors. This is due to the limitation of data, which we have only aggregate data. We point out this concern as the main limitation in this study. (Line 251-259)
we did also suggest for future research at the conclusion section (Line 274-279)
- The use of GEE with Poisson distribution is appropriate in theory, but model diagnostics, assumptions (e.g., overdispersion), and robustness checks are missing.
Response: Thank you very much for the utmost comments. We do agree. After checking it was overdispersion. We did compare the QIC between the Poisson and Negative binomial, the QIC shows extensively large when applied Poisson and smaller in Negative binomial; therefore, we did re-analysis, fitting the model with negative binomial. This was emphasized in the statistical analysis section. We also updated the results which show non-significant association. (Line 144-153)
- The variable “time” is treated linearly across 7 years; however, no information is given on how time trends or external health system changes (e.g., COVID-19) may have affected screening behavior.
Response: Thank you very much, we do plot the average screening number to see the trend change across 7 years as indicated in Figure 1. We do also add a table to compare the average screening number as indicated in Table 2.
- The interpretation of coefficients (e.g., -0.225) lacks practical context. How many fewer screenings does this represent in real terms?
Response: Thank you very much we revised the magnitude, reporting it as rate ratio as also suggested by other reviewers. Furthermore, we do report mean number of screened individuals for diabetes and hypertension (Table 2) in order to better represent the real terms.
- Claims such as “twice as low” are vague and imprecise. Use percentages or rate ratios.
Response: Thank you very much. We have revised it and reported the rate ratio.
- The presentation would benefit from graphical trends over time for T-PCU and NT-PCU, which would visually convey the temporal pattern more effectively.
Response: Thank you very much, we do plot the average screening number to see the trend change across 7 years as indicated in Figure 1.
11 The discussion is speculative in parts and lacks depth in comparing findings with international literature. For example:
- How do the findings compare with decentralization experiences in other LMICs?
Response: Thank you very much for your comments. Previous study explores the impact based on governance and financial. In Thailand, there is one that investigates the impact of transfer policy on dental health come. However, we did try as much as possible with the paper focus on health services in the discussion.
- Could the reduced screening be partially due to non-policy externalities, such as urban migration or shifting patient preferences?
Response: Thank you very much for your comments. We did not explore these issues. The data retrieved from NHSO, which focus only Thai population who are under UHC. Therefore, we could not explore the issue beyond the available of the data.
12 The role of VHV is mentioned but not empirically assessed. If VHV engagement is a hypothesized mediating factor, that should be explicitly modeled or at least flagged for future research.
Response: Thank you so much. We have marked this point for future research. (225-226)
13 The manuscript contains numerous grammatical, syntactic, and idiomatic errors. Some examples:
- “...T-PCU has twice as low screening...” → unclear and ungrammatical.
- “...was concluded, for they are in different sectors.” → confusing phrasing.
- “The screening was decreasing twice...” → needs rewording.
- The abstract is repetitive and grammatically weak. Avoid redundancy and clarify the methodology/results with concise, polished language.
Response: These have revised, and we plan to apply for the English grammar checking by the journal.
14 The introduction includes policy background but lacks theoretical framing. Include references to relevant health policy or systems frameworks (e.g., WHO's health system building blocks).
Response: We have added this framing into the introduction as we revised the whole introduction. We have revised the introduction and tailored the rationale, which begins with the 6-building box, primary health care is part of the block which major core in health services, linked to the primary care unit and its responsibility. We also point out there were many services, but the most common one is screening for NCD (DM and HT).
While the topic is important and the data potentially valuable, substantial work is needed to bring this manuscript up to the publication standards of a peer-reviewed international journal. I look forward to reviewing a revised version that addresses the above concerns in detail.
Please note that based on our new results. Therefore, we change the title to "No Impact of Primary Care Unit on Screening Services in Thailand: Transferred to the Local Administrative Organization Policy"
Reviewer 3 Report
Comments and Suggestions for Authors
This manuscript provides the impact of transferring Primary Care Units in Thailand from the Ministry of Public Health to Local Administrative Organizations and mainly focuses on screening services for diabetes mellitus (DM) and hypertension (HT). The main finding is that T-PCUs have significantly lower screening rates for DM and HT than NT-PCUs, which is a difference that grows over time. The manuscript is of significant importance. However, I find these following short comings, which if addressed, can enhance the quality of the manuscript.
- There are many errors in the abstract. A thorough rewrite is essential.
- Screening numbers are based on NHSO claim data, which may not fully capture all screening activities if some are unreported or conducted outside the claim system.
- Matching is by population and province, but there is no adjustment for other potentially relevant factors such as funding per capita, rural/urban status, accessibility etc.
- There are several aspects of awkward phrasing, typographical errors, and grammatical mistakes (e.g., “T-PCU has twice as low screening for DM (and HT than NT-PCU”). Authors should proof check for grammar and typos.
- Tables could be enhanced with clearer legends, more descriptive titles, and graphical summaries.
Thorough grammar and typo check across the complete manuscript is required.
Author Response
Response to reviewer
Thank you very much for your fruitful comments. We revised as much as possible. We hope that it will satisfy the reviewer. However, please let us know if any need to be edited.
Reviewer 3
There are many errors in the abstract. A thorough rewrite is essential.
Response: Thank you very much. We do agree. We have revised the whole abstract as indicated in the manuscript.
2. Screening numbers are based on NHSO claim data, which may not fully capture all screening activities if some are unreported or conducted outside the claim system.
Response: Thank you very much for pointing out this. Based on the availability and completeness of the data, we can’t include other screening activities.
3. Matching is by population and province, but there is no adjustment for other potentially relevant factors such as funding per capita, rural/urban status, accessibility etc.
Response: Thank you very much for your comments. We selected just these 15-PCU with the purpose of reducing the confounding of the instant transition period and also the available of the data. We did revise and emphasize it as follows
Thank you very much for your comments. We have extended our justification in the 2.2 Population and sampling and 2.3 Data source section as follows
“2.2. Population and sampling
The population were NT-PCU and T-PCU in of Thailand, total was 9836 PCU. However, in this study, the T-PCU was the PCUs that have been moved to LAO between 2007 to 2022, in which there were total of 58 T-PCU. In this study, the sample were included as follows the inclusion criteria of
1 transferred between 2007 to 2012
2 complete screening data
These criteria were used to ensure that these facilities had already undergone the transition period and organizational adjustments following the transfer policy. Therefore, the total of 15 T-PCU were selected.
2.3. Data source
To increase statistical power in this study, this study therefore purposively recruits the 15 T-HPH sample of 45 NT-HPH matched to their population within the same prov-inces. Hypertension and diabetes were defined according to ICD-10 code of E10 to E14 and I10 to I15. The hospital id, number of screenings of hypertension, and diabetes re-trieved from the National Health Security Office data-based between 2017 to 2023.
The outcome of this study is screening. Screening number was determined as number of people who have been screened in their registered PCU.”
(Line 120-138)
Furthermore, we totally agreed with reviewer concern of non-adjusting with rural/urban, staffing, geographic disparities factors and so on. As for the limitation of data, we have only aggregate data and few variables are available, so we could not take any covariates into consideration. However, we did put this issue in limitations.
4. There are several aspects of awkward phrasing, typographical errors, and grammatical mistakes (e.g., “T-PCU has twice as low screening for DM (and HT than NT-PCU”). Authors should proof check for grammar and typos.
Response: We apologize for not having gone through the grammar checking. We have revised and will proceed with the journal service grammar checking.
5. Tables could be enhanced with clearer legends, more descriptive titles, and graphical summaries.
Response: Thank you very much for your suggestions. We have revised the Table, added clearer legends, footnote and provided the figure to better understanding.
Please note that based on our new results. Therefore, we change the title to "No Impact of Primary Care Unit on Screening Services in Thailand: Transferred to the Local Administrative Organization Policy"
Reviewer 4 Report
Comments and Suggestions for Authors
The manuscript explores an important policy issue related to the decentralization of primary healthcare services in Thailand and aims to assess the effects of transferring primary care units to local administrative structures. The topic is timely and relevant, especially given the increasing interest in evaluating health system reforms and their implications for service delivery. However, the current version of the manuscript requires major revisions in order to reach the standard expected for academic publication.
The introduction addresses the broader context well but does not clearly present a specific research hypothesis or the precise gap in the literature the study aims to fill.
Some background elements, such as the historical and legislative aspects of decentralization, could be shortened or restructured to keep the focus on the research problem. The materials and methods need major revisions.
It remains unclear how the comparison groups (transferred vs non-transferred PCUs) were matched and whether any confounding variables were accounted for. Important details such as inclusion criteria, sample selection rationale, and limitations of the data source are either missing or insufficiently described.
Although the use of a GEE model is appropriate, the justification and model specification require further clarification. Also, the relatively small sample size raises questions about statistical power and generalizability.
The work has a clear and precise scientific style but not excellent graphic layout. The results are presented primarily in tables but would benefit from graphical illustrations such as trend lines or year-by-year comparisons. This would help visualize the differences over time more intuitively. Additionally, the explanation of statistical coefficients is not always clear, particularly for readers who are not familiar with log-linear regression models.
My recommendation is to divide the result and discussion sections into chapters for easier understanding. Some interpretations are speculative or based on assumptions not directly supported by the data.
The discussion would be stronger if it integrated more international literature for comparison and clearly separated interpretation, limitations, and practical implications. The potential policy relevance of the findings deserves more attention.
Revised and expanded the conclusion section. It should clearly reflect the findings and their potential significance, particularly for future monitoring and policy decisions. The manuscript includes relatively few references, many of which are local and not easily accessible to an international audience. Expanding the reference list with more international sources and recent systematic reviews.
Author Response
Response to reviewer
Thank you very much for your fruitful comments. We revised as much as possible. We hope that it will satisfy the reviewer. However, please let us know if any need to be edited.
Reviewer 4
The manuscript explores an important policy issue related to the decentralization of primary healthcare services in Thailand and aims to assess the effects of transferring primary care units to local administrative structures. The topic is timely and relevant, especially given the increasing interest in evaluating health system reforms and their implications for service delivery. However, the current version of the manuscript requires major revisions in order to reach the standard expected for academic publication.
1 The introduction addresses the broader context well but does not clearly present a specific research hypothesis or the precise gap in the literature the study aims to fill.
Response: Thank you very much for your comments. We totally agree. We have revised the introduction and provided the research gap and specific hypothesis in the introduction.
2 Some background elements, such as the historical and legislative aspects of decentralization, could be shortened or restructured to keep the focus on the research problem. The materials and methods need major revisions.
Response: We have shortened the legislative part and re-ordered the rational.
3 It remains unclear how the comparison groups (transferred vs non-transferred PCUs) were matched and whether any confounding variables were accounted for. Important details such as inclusion criteria, sample selection rationale, and limitations of the data source are either missing or insufficiently described.
Response: Thank you very much for your comments. We selected just these 15-PCU with the purpose of reducing the confounding of the instant transition period and also the available of the data. We did revise and emphasize it as follows
Thank you very much for your comments. We selected just these 15-PCU with the purpose of reducing the confounding of the instant transition period and also the available of the data. We did revise and emphasize it as follows
Thank you very much for your comments. We have extended our justification in the 2.2 Population and sampling and 2.3 Data source section as follows
“2.2. Population and sampling
The population were NT-PCU and T-PCU in of Thailand, total was 9836 PCU. However, in this study, the T-PCU was the PCUs that have been moved to LAO between 2007 to 2022, in which there were total of 58 T-PCU. In this study, the sample were included as follows the inclusion criteria of
1 transferred between 2007 to 2012
2 complete screening data
These criteria were used to ensure that these facilities had already undergone the transition period and organizational adjustments following the transfer policy. Therefore, the total of 15 T-PCU were selected.
2.3. Data source
To increase statistical power in this study, this study therefore purposively recruits the 15 T-HPH sample of 45 NT-HPH matched to their population within the same prov-inces. Hypertension and diabetes were defined according to ICD-10 code of E10 to E14 and I10 to I15. The hospital id, number of screenings of hypertension, and diabetes re-trieved from the National Health Security Office data-based between 2017 to 2023.
The outcome of this study is screening. Screening number was determined as number of people who have been screened in their registered PCU.”
(Line 120-138)
Furthermore, we totally agreed with reviewer concern of non-adjusting with rural/urban, staffing, geographic disparities factors and so on. As for the limitation of data, we have only aggregate data and few variables are available, so we could not take any covariates into consideration. However, we did put this issue in limitations.
- Although the use of a GEE model is appropriate, the justification and model specification require further clarification. Also, the relatively small sample size raises questions about statistical power and generalizability.
Response: Thank you very much for the utmost comments. We do agree. Baaed on your comments and other reviewers, after checking the overdispersion when applied Poisson. It was overdispersion. Hence, the negative binomial has been applied. We did compare the QIC between the Poisson and Negative binomial, the QIC shows extensively large when applied Poisson and smaller in Negative binomial; therefore, we did re-analysis, fitting the model with negative binomial. This was emphasized in the statistical analysis section. We also updated the results which show non-significant association. (Line 139-153)
As of the purpose T-PCU that has just only 15-PCU available data therefore we match with 1:3 ratio to improve the power. Regarding the power of test diagnostic, we spent time finding the way to calculate it, but we sincerely state that we can’t find the way to perform the power of GEE when applying with count data. However, we could see the 95%CI that demonstrate not too wide CI which indicates more precise estimates, often due to larger sample size or less variability.
5 The work has a clear and precise scientific style but not an excellent graphic layout. The results are presented primarily in tables but would benefit from graphical illustrations such as trend lines or year-by-year comparisons. This would help visualize the differences over time more intuitively. Additionally, the explanation of statistical coefficients is not always clear, particularly for readers who are not familiar with log-linear regression models.
Response: Thank you very much for the comments. Thank you very much we revised the magnitude, reporting it as rate ratio as also suggested by other reviewers. Furthermore, we do report mean number of screened individuals for diabetes and hypertension to better represent the real terms.
6 My recommendation is to divide the result and discussion sections into chapters for easier understanding. Some interpretations are speculative or based on assumptions not directly supported by the data.
Response: The results and discussion were revised accordingly.
7 The discussion would be stronger if it integrated more international literature for comparison and clearly separated interpretation, limitations, and practical implications. The potential policy relevance of the findings deserves more attention.
Response: Thank you very much for your suggestions. We have review and discuss with internation literature.
8 Revised and expanded the conclusion section. It should clearly reflect the findings and their potential significance, particularly for future monitoring and policy decisions. The manuscript includes relatively few references, many of which are local and not easily accessible to an international audience. Expanding the reference list with more international sources and recent systematic reviews.
Response: Thank you very much for your suggestions. Based on our new analysis result, we have revised the conclusion accordingly.
Please note that based on our new results. Therefore, we change the title to "No Impact of Primary Care Unit on Screening Services in Thailand: Transferred to the Local Administrative Organization Policy"
Round 2
Reviewer 2 Report
Comments and Suggestions for Authors
Although the authors have attempted to revise and enhance the manuscript, several issues remain unaddressed. I would like to mention the major points.
- There are many absolute statements in the articles, but without any references/citations. For example, “As a study in Pakistan, noted that despite increased provincial health allocations, a lack of coordination between provincial and central level impeded effective implementation".
- Major English mistakes, like no capitalisation in "1. transferred between 2007 to 2012 124 2. complete screening data".
- The results section is still in very poor shape. The authors need to revisit the descriptive part of the results.
Author Response
Thank you very much for your fruitful comments. We revised as much as possible. We hope that it will satisfy the reviewer. However, please let us know if any need to be edited.
Reviewer 1
Although the authors have attempted to revise and enhance the manuscript, several issues remain unaddressed. I would like to mention the major points.
- There are many absolute statements in the articles, but without any references/citations. For example, “As a study in Pakistan, noted that despite increased provincial health allocations, a lack of coordination between provincial and central level impeded effective implementation".
Response: Thank you very much for your comments. We have revisited the manuscript. We have cited the study in Pakistan as reference [28] “Zaidi SA, Bigdeli M, Langlois EV, Riaz A, Orr DW, Idrees N, et al. Health systems change after decentralisation: progress, challenges and dynamics in Pakistan. BMJ Glob Health. 2019 Jan 30;4(1)” (Line 371-372)
- Major English mistakes, like no capitalisation in "1. transferred between 2007 to 2012 124 2. complete screening data".
Response: Thank you very much. We have revised as “Transferred between 2007 to 2012. Complete screening data” (Line 126-127). We have corrected English grammar in this version. We have corrected the English grammar in this version.
- The results section is still in very poor shape. The authors need to revisit the descriptive part of the results.
Response: Thank you very much for your comments. We have revised and polished the descriptive part of the results (Lines 159–175). Additionally, we have updated Figure 1 to enhance its visual clarity.
Reviewer 4 Report
Comments and Suggestions for Authors
The manuscript has been thoroughly revised. I recommend acceptance.
Author Response
The manuscript has been thoroughly revised. I recommend acceptance.
Response: Thank you very much.